# Healthcare resilience: a meta-narrative systematic review and synthesis of reviews

Mark Z Y Tan ,[1] Gabrielle Prager,[2] Andrew McClelland,[3] Paul Dark[1,4]

[1]Humanitarian and Conflict Response Institute, The University of Manchester, Manchester, UK
[2]Department of International Health, Johns Hopkins University, Baltimore, Maryland, USA
[3]Alliance Manchester Business School, The University of Manchester, Manchester, UK
[4]Clinical Research Network, National Institute for Health and Care Research, London, UK

**Correspondence to**
Dr Mark Z Y Tan;
mark.tan.zy@gmail.com

## ABSTRACT

**Objectives** The COVID-19 pandemic has tested global healthcare resilience. Many countries previously considered 'resilient' have performed poorly. Available organisational and system frameworks tend to be context-dependent and focus heavily on physical capacities. This study aims to explore and synthesise evidence about healthcare resilience and present a unified framework for future resilience-building.

**Design** Systematic review and synthesis of reviews using a meta-narrative approach.

**Setting** Healthcare organisations and systems.

**Primary and secondary outcome measures** Definitions, concepts and measures of healthcare resilience. We used thematic analysis across included reviews to summarise evidence on healthcare resilience.

**Results** The main paradigms within healthcare resilience include global health, disaster risk reduction, emergency management, patient safety and public health. Definitions of healthcare resilience recognise various hierarchical levels: individual (micro), facility or organisation (meso), health system (macro) and planetary or international (meta). There has been a shift from a focus on mainly disasters and crises, to an 'all-hazards' approach to resilience. Attempts to measure resilience have met with limited success. We analysed key concepts to build a framework for healthcare resilience containing pre-event, intra-event, post-event and trans-event domains. Alongside, we synthesise a definition which dovetails with our framework.

**Conclusion** Resilience increasingly takes an all-hazards approach and a process-oriented perspective. There is increasing recognition of the relational aspects of resilience. Few frameworks incorporate these, and they are difficult to capture within measurement systems. We need to understand how resilience works across hierarchical levels, and how competing priorities may affect overall resilience. Understanding these will underpin interdisciplinary, cross-sectoral and multi-level approaches to healthcare resilience for the future.

**PROSPERO registration number** CRD42022314729.

## STRENGTHS AND LIMITATIONS OF THIS STUDY

⇒ This is the first systematic review of reviews done on healthcare resilience, at organisational and system levels.
⇒ A meta-narrative approach is used, which is particularly suitable for complex topics spanning across disciplines and hierarchical levels.
⇒ Our analysis and synthesis allow for an interdisciplinary, cross-sectoral and multi-level framework for healthcare resilience.
⇒ Individual (or micro-level) resilience is not included in this review.
⇒ A review of reviews provides a broad overview but does not encompass all details or knowledge about the topic.

measures,[2] healthcare worker burnout[3] to degrees of community trust,[4] and from international research cooperation[5] to vaccine tribalism and hesitancy.[6] Our previous understanding of healthcare resilience requires re-evaluation.

To practically respond to the whole-of-society challenges in light of COVID-19, we need evidence that effectively integrates knowledge across these disciplines. This need has been highlighted by several reviews.[7 8] The meta-narrative approach offers such a method. It has been used to track several disciplines converging within a complex field, and is useful for making sense of such complex, heterogenous, and disparate data across disciplines and sectors.[9 10] It thus forms an interdisciplinary and cross-sectoral approach to healthcare resilience.

Recognition that resilience at individual, organisational and system levels affect each other has resulted in some early multi-level empirical research protocols.[11 12] Yet, many studies tend to focus on single hierarchal levels,[13–15] within set paradigms.[16 17] For example, literature on psychological resilience tends to focus on acute interventions during or after

## BACKGROUND

The COVID-19 pandemic has tested the resilience of healthcare across the globe. Previously thriving healthcare systems have struggled with multiple aspects; from hospital capacity[1] to infection control

crisis.[18] These seldom consider wider determinants at organisational or system level which affect individual resilience.[3] Similarly, some disaster risk reduction (DRR) frameworks have put little emphasis on recovery after crisis,[19] which has been highlighted as an important factor to build resilience after COVID-19.[20] Therefore, in addition to interdisciplinary and cross-sectoral evidence, multi-level knowledge must also be consolidated.

Given the sheer volume of empirical literature available on healthcare resilience, a multi-level review was impractical. Instead, reviews of reviews are particularly useful for gaining a broad overview of complex topics and systems with many dependencies. For example, it has helped to better understand policy and practice priorities in climate sciences,[21] and to unpick COVID-19 vaccine hesitancy.[22] Both are highly complex and current topics involving multiple sectors, disciplines and hierarchical levels.

We therefore undertook a systematic review of reviews using a meta-narrative approach to present an overview and critical look at healthcare resilience. It highlights lessons learnt from the COVID-19 pandemic and synthesises evidence towards an overarching framework. The research aims to:

► Explore the definitions, measures and concepts of healthcare resilience.
► Synthesise evidence from the exploration and analysis towards a broad overview of the topic.
► Present an interdisciplinary, cross-sectoral and multi-level framework from analysed data.

## METHODS
### Theoretical approach

We undertook a systematic literature review of reviews using a meta-narrative approach.[23 24 25] We used RAMESES guidelines for reporting meta-narrative reviews.[24] Preferred Reporting Items for Systematic Reviews and Meta-Analyses (PRISMA) guidelines were followed.[26]

The principles of the meta-narrative review are pragmatism, pluralism, historicity, contestation, reflexivity and peer review. This approach was adopted for several reasons. First, the field has been informed by many research traditions. Second, there exists historical agendas which have shifted the focus of resilience at various hierarchical levels (historicity). Third, there are sometimes contrasting findings about resilience (pluralism and contestation). Fourth, an interdisciplinary overview may be useful for policymakers, practitioners and academics (pragmatism). Such an approach helps to extract seemingly heterogenous data (peer review). Finally, the final search strategy was informed by some initial searches and snowball sampling, reflecting a reflexive protocol which was adapted according to findings.

### Search strategy

A systematic keyword search was conducted in Scopus (Elsevier), Web of Science (Clarivate), PubMed (NIH) and Global Index Medicus (WHO) (figure 1). We included alternative spellings and synonyms. Inclusion criteria included reviews related to healthcare organisations and systems. We included records from 2008 to November 2022. The search was performed in January 2022 and repeated in November 2022. 2008 corresponded to the

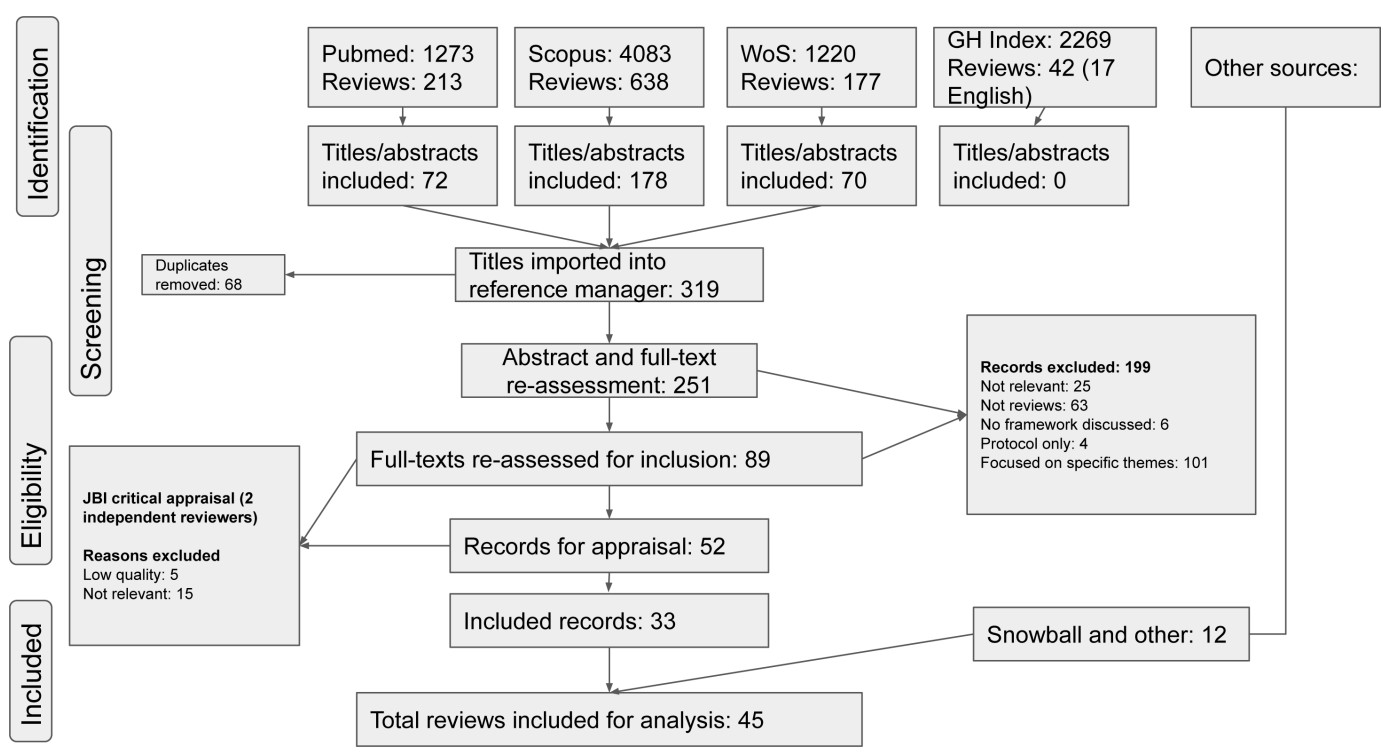

**Figure 1** Search strategy.

global financial crisis, which resulted in an increase in resilience literature.[27] Resilience is dynamic.[28] Concepts from earlier events may not possess the same relevance as more recent articles. This is especially the case since COVID-19. Reviews which were non-healthcare, focused on individual, psychological or community resilience alone, or concerned with specific diseases outside of epidemic contexts were excluded. We also excluded those focused on infection control measures only.[29] There were many reviews on individual psychological resilience, particularly acute interventions to mental health crises. While we recognised the importance of individuals that make up organisations and systems, we wanted to focus on how resilience is understood at other hierarchical levels, so we excluded these reviews.

### Study selection

Records were managed with Endnote 20 (Clarivate Analytics). Two reviewers independently selected papers. This was done for several reasons. First, there was a wide heterogeneity of studies and papers. Second, some reviews refer to cross-sectoral interactions but did not deal with healthcare. Third, the quality of reviews varied significantly. As a result, having independent reviewers (peer review) minimises personal bias and ensures there is agreement about selection criteria.[30 31] Screening was done using a modified two-stage process. Title screening was insufficient as a stage on its own due to lack of information about the actual review, or only using 'resilience' as a tangential keyword. Both reviewers thus performed a title/abstract screening independently, followed by abstract/full-text screening and discussion around inclusion and exclusion criteria. Additional reviewers were agreed a priori should disagreements between the two primary reviewers occur. The final list of included papers was supplemented by snowball sampling and hand searching. This was done by searching through relevant references from selected reviews and discussions with other interdisciplinary team members. Because of the volume of results (figure 2), it was impractical to focus on empirical studies, or we would have had to confine the search to a single hierarchical level. Therefore, we included reviews only. This may reduce details and resolution, but it allows a summary of knowledge across several reviews and disciplines.

### Quality appraisal

Quality appraisal of articles was conducted using the Joanna Briggs Institute (JBI) checklist for systematic reviews.[32] While there are no specific quality appraisal tools for systematic reviews of reviews, the JBI checklist formed a best fit compared with several other tools (online supplemental appendix 1).

### Data extraction and analysis

Using an inductive approach, three-tiered coding of the text was conducted around the three foci of definition, measures and concepts[33] (online supplemental appendix 1). We used Atlas.ti qualitative software and data extraction tables. We examined prominent definitions and how they have been informed by different research disciplines. We then collated definitions from the reviews and analysed the most frequently occurring words (online supplemental appendix 2). These results were categorised and synthesised with the concepts of resilience to produce a definition which acknowledges past research, links current knowledge and spans hierarchical levels. Concepts were coded from reviews as well as from the frameworks presented in papers (online supplemental appendix 3).

# Search Strategy

## Scopus (Elsevier)

(TITLE-ABS-KEY ("resilien*"OR"strengthening") AND TITLE-ABS-KEY ("system"OR"organi*ation") AND TITLE-ABS-KEY ("health"OR"hospital"OR"healthcare") AND TITLE-ABS-KEY ("crisis"OR"disaster"OR"epidemic"OR"pandemic"))

## Web of Science (Clarivate)

("resilience" OR "resilient" OR "strengthen") AND ("health") AND ("organisation" OR "systems") AND ("crisis" OR "disaster" OR "epidemic" OR "pandemic")

## PubMed (NIH)

(((("resilien*"OR"strengthening") AND ("system"OR"organisation")) AND (health)) AND ("crisis"OR"disaster"OR"epidemic"OR"pandemic")

## Global Index Medicus (WHO)

(tw:(Resilien* OR strengthening)) AND (tw:(System OR Organisation))

## Inclusion Criteria

- 2008 - 2022
- Scholarly articles (reviews) published in peer-reviewed academic databases
- Related to healthcare organisations and systems

## Exclusion Criteria

- Non-healthcare
- Focused on individual, psychological or community resilience
- Concerned with a particular disease or health condition outside of epidemic/pandemic context
- Non-English

**Figure 2** Preferred Reporting Items for Systematic Reviews and Meta-Analyses diagram.

## Patient and public involvement

There was no patient or public involvement in this research.

## RESULTS

Four databases yielded 8845 records, of which 1070 were reviews. First-stage screening yielded 319 records, of which 68 were duplicates. Two hundred and fifty-one records were re-screened and discussed. The remaining 52 records underwent quality appraisal. This resulted in 33 records from the search which were analysed. Five reviews were excluded for low quality. They lacked details about search strategy, inclusion criteria, data analyses, or indeed clear research questions. Other reviews were not relevant according to inclusion and exclusion criteria. Snowball sampling yielded 12 records for analysis. These include reviews which discussed resilience frameworks. The PRISMA diagram summarises the selection process (figure 2).

## Definitions of healthcare resilience

Most reviews took their definition of healthcare resilience from previous papers. Three reviews contained lists of definitions from other studies.[17 34 35] Another three reviews presented narrative collections and analyses of definitions.[15 36 37]

Resilience tends to be defined based on paradigms within which it operates.[35–37] For example, within the engineering paradigm it is concerned with the ability of a structure or material to return to its original state.[16] However, within business and ecological systems, it includes the ability for an organisation or system to capitalise or improve after a shock.[37–39] A prominent definition of healthcare resilience is from Hollnagel *et al* as 'the ability of the health care system (a clinic, a ward, a hospital, a county) to adjust its functioning prior to, during, or following events, and thereby sustain required operations under both expected and unexpected conditions'.[40 41] While this definition continues to be refined, it provides a frame to highlight several aspects which emerged from our word occurrence analysis too.

First is the recognition of different hierarchical levels of the healthcare system[41 42] (figure 3). There is consensus of these as micro-level (referring to individual), meso-level (hospital or organisation), macro-level (region or country).[41 42] Several nuances continue to evade neat classification. For example, the team level does not neatly fit into micro-level or meso-level,[43] and a community has been considered both meso-level and macro-level.[36 37] There is also emergence of a meta-level of healthcare resilience. This level focuses mainly on the climate crisis as a determinant of health and how it threatens healthcare resilience and exacerbates inequalities.[8 44–46] This transcends national boundaries and so must be considered on a meta-level.

Second, resilience spans pre-event, intra-event and post-event. This appreciates contributions from different paradigms. DRR studies are mostly concerned with pre-event aspects, such as preparing for shocks thereby minimising disturbance to a system.[13 47–51] Emergency management (EM) studies examine aspects of how organisations adapt during a crisis (intra-event), and how they may learn from

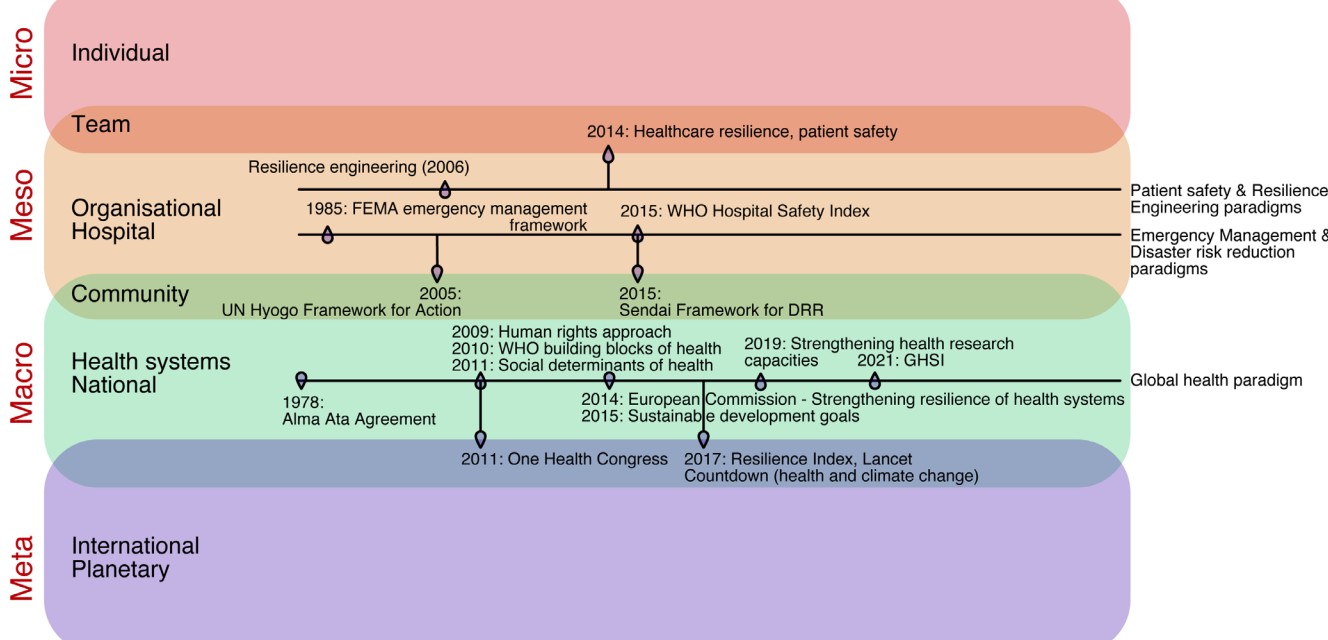

**Figure 3** Healthcare resilience: hierarchical levels and selected paradigm timelines with historical highlights. DRR, disaster risk reduction; FEMA, Federal Emergency Management Agency; GHSI, Global Health Security Index.

it.[17 52] Some definitions do not include recovery, while others emphasise that rapidity of recovery back to steady state is a key feature of resilience.[16 36 49 53]

Third, both expected and unexpected events are considered. DRR and EM paradigms are concerned with disasters and unexpected crises, which suggest that resilience is only manifest when a sufficiently large shock is applied to the system.[34] Patient safety paradigms seek to minimise unintended harm to patients, whether in crisis situations, or within normal operational stressors in healthcare.[7 35 37 54] Recently, even DRR and EM reviews have adopted such an 'all-hazards' approach to resilience.[1 46]

Hollnagel *et al*'s definition of healthcare resilience, though widely used, does not specifically include recovery or review.[40] Instead, he refers to these concepts as 'learn' within his related Resilience Analysis Grid (RAG).[55] The resilience potentials in RAG are anticipate, respond, monitor and learn. Many other definitions map well onto Hollnagel's definition and his four potentials. There are benefits in linking up definitions with 'potentials' or concepts of resilience. It provides convergence in a complex field, which helps to develop a consistent approach across hierarchical levels. This may promote

understanding between policymakers and healthcare practitioners (including leaders and managers). In turn, it may lead to more unified approaches across discipline and hierarchical boundaries.

We performed word occurrence analysis from the included reviews, followed by tiered coding into themes, with iterative refinement based on the analysis of concepts of resilience (online supplemental appendix 2). The result is a synthesised definition which complements previous work, spans across disciplines and dovetails with concepts explored in the next section. Our definition of healthcare resilience is 'the ability of healthcare workers, organisations or systems to (a) prepare for and prevent, (b) absorb and adapt to maintain structure and essential functions, (c) recover and review from crises, shocks or stressors' (figure 4).

### Concepts of healthcare resilience

Concepts may refer to practices or features which promote or hinder resilience, or indicators of resilience. Groups of concepts are often collated in frameworks, or set as best fit into established frameworks (eg, WHO building blocks of health). Frameworks are useful to systematically consider various aspects of healthcare resilience.[46 56] We

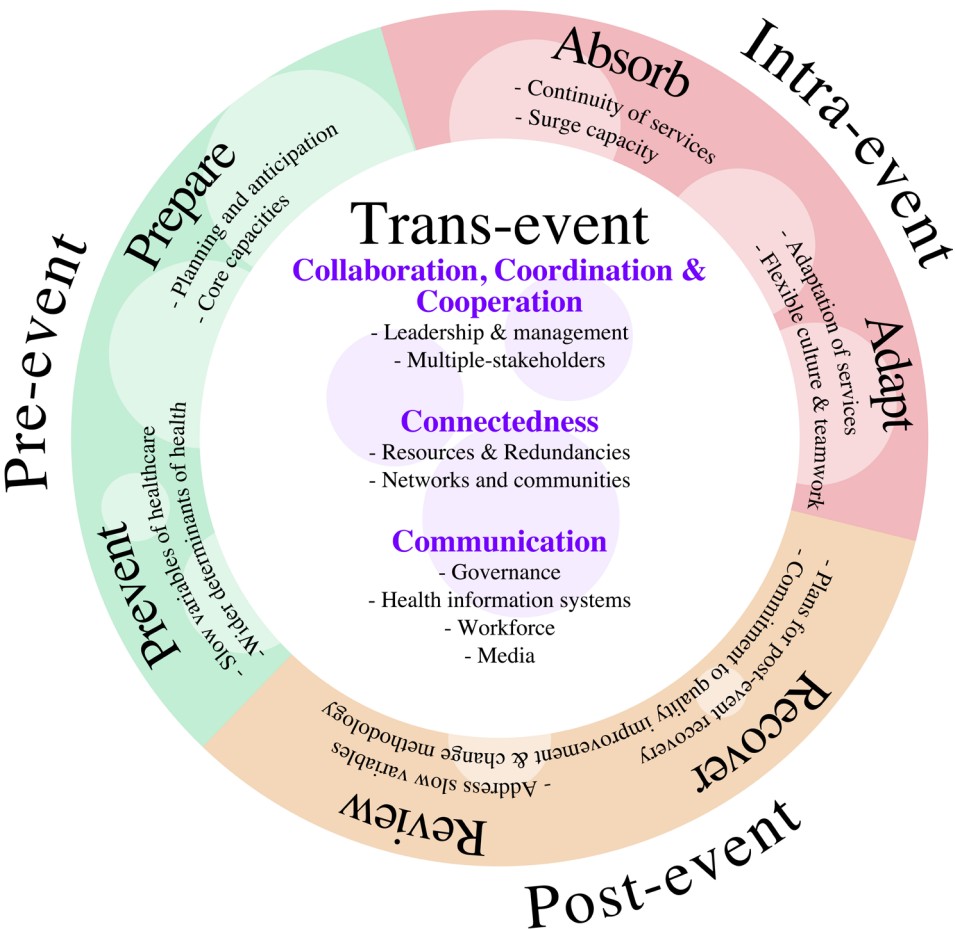

**Figure 4** An interdisciplinary, cross-sectoral, and multi-level definition and conceptual framework for healthcare resilience. Defining features are listed as headings within the pre-event, intra-event, post-event and trans-event time domains.

highlight key concepts at macro-level and meso-level, paying attention to lessons learnt from the COVID-19 pandemic. We then present our framework of concepts learnt from this review (figure 5), which dovetail with our definition (figure 4).

## Physical capacities

Within the DRR and EM paradigms, strengthening of existing capacities feature most prominently. This includes adequate workforce,[57] equipment,[42 58] financial resources[4] and services[4 57] to ensure maintenance of essential functions during a crisis. From the COVID-19 pandemic, appropriate distribution of physical capacities during various stages of a crisis are equally important. For example, changing shift patterns of emergency staff for increased demand intra-event, but redistributing staff to address longer-term care needs post-event.[59] This adopts a longitudinal approach to resilience past the acute crisis and helps to address predicted shortages. As such, it may then prevent mere shifting of crisis from one organisation to another.

## Multi-level approaches

Several reviews have highlighted the multi-level approaches required to build resilience.[12 41 60–63] A small number of empirical studies are currently underway to understand how this works in practice.[11 64] Multi-level approaches recognise that while previous boundaries help understand specific details about how individual hierarchical levels pursue resilience, they are insufficient in representing the interconnections between levels. As an example, while improvisation and coping were seen as resilient concepts at the micro-level, healthcare workers' ability to continue working was underpinned by the availability of resources and protective equipment provided by the organisation and their leaders (meso-level), as well as recovery efforts introduced after the crisis at meso-level and macro-level.[4 42 58] Further work is needed to better understand the interplay and dynamics between the hierarchical levels.

## Recovery and review

The most appropriate method of quality improvement to ensure learning after a crisis depends on context. It is more important for organisations to demonstrate consistency and commitment to change methodology, rather than any specific tool or method.[65] This is termed 'review' in our framework. Within the patient safety paradigm, two prominent and complementary concepts are safety1 and safety2 approaches. The safety1 approach attempts to get to the root cause of adverse events in healthcare, while the safety2 approach recognises that success occurs more frequently than adverse events and therefore lessons can be learnt from such practices too.[40]

Recovery is mentioned in DRR/EM literature,[16 53 66] but less so in patient safety literature. This could be contextual. In DRR/EM, crises tend to be larger in scale and affect more people, organisations and sectors outside of healthcare. Patient safety tends to focus on teams within clinical emergency scenarios or normal operational circumstances. DRR/EM paradigms therefore require a more concerted recovery effort. Rapidity of recovery has been cited as a feature of resilience, but there is no benchmark. Extent of recovery to previous function has been explored, but this does not appreciate the occasional change in function of a system following a large crisis.[14 36] Several reviews have emphasised the importance of having plans for recovery within resilience frameworks, but do not elaborate on details.[42 57 58] Therefore, there is still much to understand about recovery, and some recent frameworks summarise key lessons learnt from COVID-19.[20]

## Relational aspects

The COVID-19 pandemic has highlighted the importance of relational aspects of resilience. A key aspect is effective leadership.[1 13 17 36 46 48 49 52 54 59–61 63 66–71] Collaborative and authentic styles which were visible and approachable seemed most favoured during crises.[1 2 4 12 17 53 54 66 69 70 72 73] Command-and-control structures have been cited, but these refer more to the flow of information rather than organisation of authority. Consequently, recent DRR reviews advocate decisions being pushed out towards the periphery, enabling more rapid adaptation to change.[16 49 52 57]

Networks beyond formal healthcare facilities bolsters resilience through the availability of physical capacities (eg, financing, staff, equipment and volunteers).[1 13 16 36 59 60 66 71 72 74] It also facilitates adaptation and planning post-event. It also facilitates post-event concepts. For example, COVID-19 clinical and vaccine trials depended on international collaboration between many disciplines and industries.[5] Recognising the importance of this concept, the recent UK Health and Care Act 2022 specifically focuses on interorganisational relationships towards the provision of integrated care. Further research will therefore be required to better understand how these relationships work in practice.[75–77]

## Interdisciplinary, cross-sectoral and multi-level framework for healthcare resilience

These concepts, along with others which emerged from analysis of the reviews, have been compiled into an interdisciplinary, cross-sectoral and multi-level framework (figures 4 and 5, online supplemental appendix 3). It spans meso-level and macro-level, which few existing frameworks do. It balances the historical focus on physical capacities with relational aspects of resilience. It incorporates lessons learnt from the COVID-19 pandemic. The framework may be used as tool to systematically consider various aspects of organisational and systems resilience.

The framework is divided into timeframes (pre-event, intra-event, post-event and trans-event). Within timeframes are themes (prepare and prevent, etc). Themes encompass several concepts, and finally, these concepts are further broken down into components (figures 4

**Interdisciplinary, Cross-sectoral, Multi-level framework for Healthcare Resilience**

| | Themes | Concepts (frequency) | Component | Example considerations |
|---|---|---|---|---|
| **Pre-event** | Prevent | Slow variables (n=8) | Healthcare access and range of services | Universal healthcare, geographical access factors, equality in access |
| | | | Acting on risks identified | Health inequalities, at risk populations |
| | | | Strengthening healthcare infrastructure | Ventilation and air-conditioning systems, integrity of buildings, additional cliincal spaces |
| | | | Healthcare expenditure and funding channels | WHO Building Blocks of Health |
| | | | Staff retention and education | From COVID19: burnout, wellbeing, development opportunities. Plan for attrition. |
| | | Wider determinants (n=10) | Social determinants of health | Water, sanitation, education, employment |
| | | | Health inequalities | From COVID19: disproportionate health outcomes for ethnic minorities and lower socioeconomic groups |
| | | | Physical and critical infrastructure | Road and access, buildings and public services |
| | | | Environmental considerations | One Health movement, planetary health |
| | Prepare | Planning & anticipation (n=12) | Disaster or crisis protocols and policies | Sendai Framework for Disaster Risk Reduction, FEMA emergency management cycle |
| | | | Strengthen community and long-term care | From COVID19: lack of workers in care settings after initial surge |
| | | | Modelling disasters | |
| | | | Strengthen interorganisational networks | Transport and supply chains, voluntary sectors, faith-based organisations |
| | | | Training and maintenance of skills | Disaster and major incident training, simulation and drills. |
| | | Core capacities (n=11) | Beds, staff, equipment, medicines | Existing resources and emergency stockpiles |
| | | | Plans for utilising surge capacities | Equipment, spaces, alternative facilities |
| **Intra-event** | Absorb | Continuity of services (n=19) | Maintain essential services | Maternity, paediatrics, emergency medicine, intensive care |
| | | | Planning and involving spontaneous volunteers | ISO 22319:2017 |
| | | | Mass-casualty triage protocols | Royal College of Emergency Medicine guidance |
| | | | Ensure access to healthcare services | Consider hard-to-reach populations |
| | | Surge capacity (n=16) | Mobilising core and surge capacities | |
| | | | Activating policies and processes for major incidents | |
| | | | Donation standards and distribution | |
| | Adapt | Adaptation of services (n=17) | Temporary reduction in elective services | From COVID19: reduction in volume of services (e.g. USA's Crisis Standards of Care) |
| | | | Temporary incident command and communication system | |
| | | | Adopting new technologies | From COVID19: telemedicine and videoconferencing |
| | | Flexible culture and teamwork (n=7) | Collective sensemaking and fast variables in healthcare | From COVID19: improvisations and workarounds characterised resilient adaptation. |
| | | | Trade-offs, workarounds and improvisation | WAI/WAD. Actuality:Potentiality ratios |
| **Post-event** | Recover | Plans for post-event recovery (n=8) | Acute psychological interventions for staff | |
| | | | Staff for increased long-term care | Predicted late surge in community, complex and long-term care |
| | | | Strategies for community recovery | Recovery for Development Framework. National Consortium for Societal Resilience |
| | Review | Commitment to quality improvement and change methodology (n=16) | Quality improvement method or framework | Evaluation reports, vulnerability analysis, risk assessments. Safety 1 & 2 |
| | | | Evaluation and data collection systems | |
| | | Address slow variables (n=8) | Organisational or system adaptation based on lessons learnt | |
| **Trans-event** | Collaboration, Coordination, Cooperation | Leadership & management (n=20, 12 respectively) | Effective leadership and trust | Open style leadership during crisis |
| | | | Crisis leadership training | |
| | | Multiple stakeholders (n= 3) | Links with surveillance organisations | Meteorological institutes, public services, public health authorities |
| | | | Cross-sector collaboration | Patient groups, manufacturers, volunteer and faith-based sector, private, non-profit |
| | | | Diverse approaches to support creativity and innovation | Kruk's Resilience Index, Principles of Socio-ecological resilience, Empirical studies of Uganda's public corporations |
| | Connectedness | Resources and redundancies (n=22) | Supply and stock management processes | |
| | | | Plans for pooling resources | From COVID19: Capacity transfers in intensive care |
| | | Networks and communities (n=16) | Integrated care | UK Health and Care Act 2022 |
| | | | Dialogue with community leaders & volunteers | |
| | | | Regional and international links and diplomatic ties | International emergency response for earthquakes and natural disasters |
| | Communication | Governance (n=5) | Ensure multiple feedback loops | |
| | | | Civil registration and statistics systems | |
| | | Health information systems (n=7) | Multi-level surveillance strategies | Public health functions, routine data collection and dissemination, data sharing and integration, early warning systems |
| | | Workforce (n=17) | Feedback & communication mechanisms across hierarchical levels | Staff forums, incident reporting |
| | | | Staff recognition | |
| | | Media (n=3) | Accuracy of information disssemination | From COVID19: false news and vaccine hesitancy |
| | | | Population trust in health system | |

**Figure 5** Details of the interdisciplinary, cross-sectoral and multi-level framework for healthcare resilience. Concepts are presented with frequency of occurrences as n-numbers. They are further divided into components, along with example considerations extracted from reviews. FEMA, Federal Emergency Management Agency.

and 5). In figure 4, the size of faint circles indicates the frequency of concepts extracted from the reviews, which are also represented as n-numbers in figure 5.

## Measures of resilience

Due to the considerable global interest in healthcare resilience, there have been attempts to develop measures or indicators for resilience. The process of pursuing resilience requires ways to determine progression. Attempts to measure resilience therefore allow comparisons and identification of potential areas for improvement. However, none of the current measures have been validated, and some widely used indices have been shown to be inaccurate during the COVID-19 pandemic.[78]

Two examples at the macro-level are the Global Health Security Index and Epidemic Preparedness Index. They both focus on infectious diseases outbreaks and contain several domains including public health and healthcare infrastructure (eg, surveillance capabilities, medical workforce), economic resources, and risk assessment and communication strategies.[2 79] As such, they are focused on the pre-event domain of crises. However, many high-income countries that scored well on these indices performed very poorly during the COVID-19 pandemic.[78] Several reasons have been cited, including lack of consideration of existing health inequalities, globalisation, societal connectedness and a bias towards physical capacities (eg, number of hospitals or beds).[78] Of note, increased globalisation suggests that 'disease control may be only as effective as practices within the poorest performing countries'.[78] This supports the need for a meta-level of resilience where global health inequalities affect the healthcare resilience of other nations.

Some reviews attempt to evaluate a country's resilience according to one or more domains of the WHO building blocks of health framework.[14 57 63 67 80] However, the framework was designed as a systematic approach to the funding decisions within healthcare, rather than to evaluate resilience.[35]

Meso-level measures tend to stem from DRR/EM paradigms, focused on healthcare facilities. Several reviews have collated evidence from empirical papers centred around earthquakes and natural disasters. Quantitative measures include physical capacities (eg, number of beds, staff, equipment), physical infrastructure (eg, structural integrity of buildings)[1] and actuality:potentiality ratios (eg, adherence to operating protocols).[36 81] Qualitative indicators include checklists containing several components important during disaster situations (eg, pre-emptive protocols and procedures, command structures, compliance to structural regulations).[16 66] These qualitative components address the intra-event domain of the resilience framework. By providing measurement tools and checklists, pre-emptive strengthening of intra-event components can help to reduce the impact of crises when they do occur.

In the patient safety paradigm, the Checklist for Assessing Institutional Resilience contains broad domains to help hospitals consider several concepts of resilience.[82] It was not designed as a set of measures but may be considered a tool for systematic thinking in patient safety. Similarly, while some have suggested that the work-as-imagined versus work-as-done (WAI/WAD) ratio (or the more general actuality:potentiality ratio) may be used as a way to improve patient safety (by minimising performance variability),[17] others highlighted that this approach may work for some simple processes, but is unlikely to result in favourable patient outcomes in unexpected or complex scenarios.[41] For example, adherence to sepsis guidelines (WAI) is audited and associated with overall better outcomes,[83] but depends on accurate diagnosis, which is itself difficult. Sepsis is frequently misdiagnosed (WAD), leading to wrong treatments, antimicrobial resistance and poor outcomes.[84 85] Therefore, WAI/WAD is not an adequate approach to measure resilience. Instead, WAI/WAD should be used as means to understand the balance struck within systems, and a concept with which to frame observations about adaptations in healthcare. This reframes resilience from an outcome-oriented approach to a process-oriented one.

Some multi-level assessments of the resilience of critical infrastructures exist. The Tiered Approach by Linkov *et al* highlights three tiers of assessments focusing on relationships between components of a system.[86] A key feature of this is the use of multiple tools for assessment at different levels and for different goals. For example, the Functional Resonance Analysis Method[87] is useful for mapping relationships between components, while indices may help to align proposed changes with the overarching goals of the system. Checklists may be useful during a crisis. Modelling can help to predict consequences, audit progress and address slow variables. As such, our framework (figures 4 and 5, online supplemental appendix 3) and the related data extraction table (online supplemental appendix 4) provide examples of resilience frameworks mapped onto the time domain of crises. There are contemporary studies which seek to better understand multi-level influences specific to healthcare resilience.[76 77]

There seems to be limited focus on the measurement of post-event and trans-event domains. While quality improvement models tend to be embedded within healthcare organisations and systems, assessment of speed of recovery and robustness of interorganisational relationships rarely feature in assessment tools. Similarly, assessment of culture and leadership is challenging.[88]

## DISCUSSION

We have employed a relatively new review methodology well-suited for complex topics informed by various research traditions. There is increasing acknowledgement of the need for interdisciplinary, cross-sectoral and multi-level approaches to healthcare resilience. Yet, its history has been one of multiple research traditions operating within specific paradigms. Global health paradigms

dominate the macro-level, while DRR and patient safety frame the meso-level.

Definitions of healthcare resilience are increasingly broad. They have evolved from highly contextualised situations towards appreciation of different hierarchical levels and from an outcome-oriented understanding towards one which is more process-oriented. With this evolution, it has become more difficult to characterise failure of resilience. Even though healthcare systems and organisations are affected by a range of stressors, they are inherently resilient as essential societal services. Recognising the interplay between hierarchical levels is vital for building resilience. For example, the Institute for Healthcare Improvement framework for improving joy in work considers meso-level and macro-level aspects (eg, participative management and an environment of psychological safety), that affect micro-level resilience, and thus, explicitly links previously distinct hierarchical levels.[89] Such multi-level frameworks rarely feature at the meso-level and macro-levels, despite healthcare workers being arguably the most important resources of healthcare systems. Several recent study protocols have been published to better understand multi-level influences in resilience, particularly in the patient safety paradigm.[11 64]

Put another way, resilience concepts at macro-level have tended to focus on physical capacities prior to COVID-19. Yet, there is increasing recognition of the importance of relational aspects of resilience, particularly across macro-level and meso-level boundaries. Some examples of these within the COVID-19 pandemic include the adoption of open leadership at meso-level and macro-levels,[58] interorganisational relationships and their realignment,[4 60] cultivating an environment of trust,[42 59] and effective communication channels at micro-level, meso-level and macro-level.[90] Open leadership styles gained more trust and enabled better understanding of WAI/WAD.[42] Alongside, effective channels of communication between healthcare workers, managers and in turn national bodies helped with rapid adaptation and identification of acute needs, from equipment and supply issues to psychological support services.[58]

While a range of tools and indicators have been developed for physical capacities and infrastructure, it is still not known how best to assess the effectiveness of relational aspects of resilience. In particular, understanding the nature of interorganisational relationships seems to stand out as a research priority.

Our framework for healthcare resilience fulfils several needs in this field. First, it brings together knowledge and observations gained from a range of disciplines, across hierarchical levels, and integrates it with lessons learnt from the COVID-19 pandemic. It acknowledges and builds on historical health agendas across geopolitical and financial contexts (figure 3). Many of these have been centred in low/middle-income countries including WHO's building blocks of health, health systems strengthening and DRR agendas (online supplemental appendix 3). As such, our framework may be applied to a wide range

of health organisations and systems. It suggests that resilience requires simultaneous consideration of a diverse set of components but does not assert a particular component as most important. Second, it emphasises both physical and relational aspects of resilience (figures 4 and 5). Relational aspects such as cross-sector networks and collaborations feature in global health agendas developed primarily for low/middle-income countries but have hitherto received less attention in the indices applied to high income countries. Thus, different components of the framework may take priority for different systems at different points of time. Finally, it dovetails with our definition and achieves some convergence in a highly heterogenous field.

There are several strengths to this study. First, a review of reviews provides a broad overview of the topic across several geopolitical contexts. There is consolidation of knowledge across a wide range of disciplines and paradigms, enabling wider appreciation of the complexities of the topic and the many ways it has been approached. Second, a meta-narrative approach helps to explore historical trends and ongoing debates within a complex field. In addition to the consolidation of knowledge in the previous point, this approach also acknowledges the differences between paradigms and broadens the understanding of resilience (figure 3, online supplemental appendix 4). Third, we have taken a multi-level approach at meso-level and macro-level, which has hitherto been rare. These three points provide a map for healthcare workers and health systems managers to better understand the contexts within which each party operates.[3] Fourth, we have highlighted several lessons learnt from the COVID-19 pandemic, which was a shock to entire healthcare systems across the world.

Several limitations persist. We do not focus on the micro-level, even though humans are undoubtedly the most important components of healthcare systems, both as staff and service-users. Empirical studies are excluded. This means that while we were able to maintain breadth, we inadvertently lost resolution. In addition, knowledge continues to be discovered and some will not have made it into reviews yet. Therefore, while this review of reviews is timely, we acknowledge that resilience is a dynamic process.[28] Several important empirical studies are underway to unpick key relational factors between hierarchical levels and how they may support or hinder adaptive capacity.[11 76] These studies should help to deepen understanding into healthcare resilience. We chose not to include grey literature due to volume of work, but this potentially excludes some important information from this review. Similarly, while the language restriction potentially imposes a selection bias, a significant proportion of included reviewed specifically focused on non-English speaking and/or low/middle-income countries. Frameworks can be considered tools for systematically approaching resilience but require adaptation towards individual situations (online supplemental appendix 4). Our framework does not provide guidance on which

components should be prioritised by which health system at which point of time. This requires contextualised analyses and aligns with the related realist approach seeking to understand 'what works, for whom, under what circumstances and how'.[24]

Future reviews should state clear research questions, adopt systematic search strategies and use robust data analyses techniques. These would maximise the quality of evidence generated. Future priorities arising from this research include the need to examine multi-level resilience with empirical studies,[11] exploring how each level's resilience contributes to other levels[3] and gaining deeper understanding of the nature and extent of interorganisational relationships and how they affect resilience.

## CONCLUSION

Our review of reviews provides a broad overview of healthcare resilience. The meta-narrative approach adopted recognises the contribution from several disciplines and paradigms within this field. We have highlighted the tendency for studies to focus on individual hierarchical levels and limited timescales, but also acknowledge the need and current desire to develop more interdisciplinary, cross-sectoral and multi-level approaches to healthcare resilience. Our framework for healthcare resilience crosses hierarchical and time boundaries, providing a broad and reflexive overview of healthcare resilience. Understanding the balances between sometimes opposing concepts, and contrasting priorities of different professional groups or hierarchies, is key to building resilience for the future.

**Contributors** Concept: MZYT, AMC, PD. Planning and design: MZYT. Screening: MZYT, GP. Additional reviewers: AMC, PD. Data collection, analysis and interpretation: MZYT, GP. Writing: MZYT. Editing: all authors. Guarantor: MZYT on behalf of all authors.

**Funding** The authors have not declared a specific grant for this research from any funding agency in the public, commercial or not-for-profit sectors.

**Competing interests** None declared.

**Patient and public involvement** Patients and/or the public were not involved in the design, or conduct, or reporting, or dissemination plans of this research.

**Patient consent for publication** Not applicable.

**Ethics approval** Not applicable.

**Provenance and peer review** Not commissioned; externally peer reviewed.

**Data availability statement** All data relevant to the study are included in the article or uploaded as supplementary information.

**ORCID iD**
Mark Z Y Tan http://orcid.org/0000-0003-3330-728X

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
