## [Reviewer comments · BMJ Open]

ARTICLE DETAILS

TITLE (PROVISIONAL)	Healthcare resilience - A meta-narrative systematic review and synthesis of reviews
AUTHORS	Tan, Mark; Prager, Gabrielle; McClelland, Andrew; Dark, Paul

VERSION 1 – REVIEW

REVIEWER	Greer, Scott University of Michigan, Health Management and Policy
REVIEW RETURNED	27-Mar-2023

GENERAL COMMENTS	This is a valuable meta-narrative review and synthesis of reviews on an important topic. It is well done and reported, transparent about decisions, and worthy of publication. It would possibly be improved by the authors' being a bit clearer about exactly how their synthetic definition arose and how it relates to the very different approaches they identified (e.g. engineering, emergency management)- what is left out in this synthesis and does it matter?
--

REVIEWER	Bal, Roland Erasmus University Rotterdam, School of Health Policy & Mangemen
REVIEW RETURNED	01-May-2023

GENERAL COMMENTS	This is a well-designed and executed review of an important topic. The authors have done a fine job in reviewing the literature and have used interesting methods in doing so--e.g. using a world clad to look at the most-used concepts in relation to healthcare resilience. The choice of focussing on organisational and system levels is well-explained and justifiable given the wealth of the literature on the topic. The definition of resilience the authors derive from the literature is also well argued. I specifically liked that the authors not only analyse and summarise the literature but also come up with suggestions for further research and feel that the turn the authors describe in the literature from more structural to more processual approaches to resilience is a valid one. The authors rightly point out that a limitation of the method they chose is that it remains rather abstract, while they point out that resilience is largely to be found in the nitty-gritty details of the relationality between layers (e.g. micro-meso-macro relations). I feel they could elaborate a bit more on how this relationally might look like and what kind of work would be needed to research this. For example, Wiig et al have pointed at the importance of 'reflexive spaces' between layers whereas others have pointed at the rol of 'dynamic protocols'.
---

REVIEWER	Haywood, Philip Centre for Health Economics Research and Evaluation (CHERE), Business My work at the OECD involves the consideration of frameworks for resilience.
REVIEW RETURNED	02-May-2023

GENERAL COMMENTS	Thank you for the opportunity to review the paper “Healthcare resilience – A meta narrative systemic review and synthesis of reviews” I enjoyed reading the paper and think it is a valuable contribution to a very fast-moving field. The authors are to be congratulated on a synthesis that was clear and highlighted several important issues. I also thought their proposed framework highlights several important issues that are valuable to conceptualising resilience. The paper synthesises reviews of health system resilience and uses the information to develop a universal framework, as well as highlighting differences between reviews in their approach to conceptualising health system resilience. The title and abstract are appropriate. I have several comments aimed at drawing out specific issues. I thought some of the material that was included in the appendices may be better placed in the main text and discussed more fully. Additionally, there were some areas I would have appreciated some more information. I thought the completeness of the search strategy was the most important area for the authors to comment upon. Major comments for which I think inclusion in the paper would be very useful:  1. Please include more formally the inclusion and exclusion criteria in the methods. One area in which I got confused was the issue of community resilience. It was included in the exclusion criteria for the review, whereas there was a discussion of the importance of the community in several areas. The distinction between inclusion and exclusion of a primarily community or socially based review was difficult to appreciate from the paper. The New Zealand Living Standards Framework was the one that most caught my attention as the paradigm listed in Appendix 4 (page 16) was community. I was not sure of exactly how this exclusion criterion was operationalised in the exclusion of specific articles. The implication was that if the assessment was of community resilience and included healthcare it was included or if not excluded. Because the review was not restricted with regard to the income of countries, this may have had implications for countries with larger voluntary or informal sectors. Additionally, although it can be inferred from appendix 3 it may be worth noting the coverage of high income health systems versus low-middle income health systems. The definition of exactly what is in the health care
--

sector may vary depending on the context.

2. Section 3. I would have appreciated **more information about the quality assessment results**. As discussed in page 7 the quality varied greatly and some guidance on what could be improved might be useful for future authors of reviews.
3. Some of the **weaknesses that may be produced from the methods used could have been made more explicit**. The lack of inclusion of the grey literature is entirely justifiable from the pre-specified methods but does leave some potential information from national and international organisations excluded, for example European Commission’s work on resilience (EU Expert Group on Health Systems Performance Assessment (HSPA), 2020), similarly with the restriction on English. It may be these are more important for the global level of the framework.
4. Similarly, it would be worth **commenting on the completeness of search strategy** and the approach used to make it complete. One specific issue noted was the use of “organisation” in quotes for the Pubmed search, alternatives could have been “organization” or the combination of “organisation” or “organization”. See the table below for the potential differences in the number of recovered articles for screening. A similar issue may occur with “crisis” and “crises”.

Table 1. Organisation or organization in search string

Pubmed search string
<pre> ((((("resilien*"OR"strengthening") AND ("system"OR"organisation") (health)) AND ("crisis"OR"disaster"OR"epidemic"OR"pandemic")) (("2008/01/01"[Date - Create] : "2022/11/01"[Date - Create])) </pre>
<pre> ((((("resilien*"OR"strengthening") AND ("system"OR"organization") (health)) AND ("crisis"OR"disaster"OR"epidemic"OR"pandemic")) (("2008/01/01"[Date - Create] : "2022/11/01"[Date - Create])) </pre>
<pre> ((((("resilien*"OR"strengthening") AND ("system"OR"organization" "organisation")) AND (health)) ("crisis"OR"disaster"OR"epidemic"OR"pandemic")) (("2008/01/01"[Date - Create] : "2022/11/01"[Date - Create])) </pre>

Note: Search conducted on the 2/05/2023 and undertaken in pubmed (<https://pubmed.ncbi.nlm.nih.gov/>)

Major comments about where additional discussion may be useful in the paper.

These are the areas where additional questions were raised as reading the paper. The ability to address these areas is dependent on space, they are presented for the authors consideration.

1. Section 3.1: The definition of healthcare resilience

was: Our definition of healthcare resilience is the ability of healthcare workers, organisations or systems to (a) prepare for and prevent, (b) absorb and adapt to maintain structure and function, (c) recover and review from crises, shocks or stressors. Please consider the following issues:

- Does essential need to be included with function? Or is it all functions?
- Is the meta, global system discussed in the article accurately described as a healthcare system? It is not inaccurate but important components of the global health architecture sit outside of health and different considerations are made at a meta level than a national or regional one (https://apps.who.int/gb/ebwha/pdf_files/WHA75/A75_20-en.pdf).

2. Section 3.3. Measurement of resilience was a key component of the paper and extremely interesting to read. The lack of inclusion of empirical studies may result in validation studies not being included but this is not an important issue given the prespecified search strategy.

There appeared to be a potential opportunity to discuss the association between measurement and the framework the authors have designed. Specifically, while the levels concept was included there was less inclusion of the time dimension of the framework (specifically the pre, intra and post). More explicit linking of this section to the framework that was developed would be useful. There were some specific areas in the themes presented on page 29 where this could have been undertaken.

As the authors have discussed measuring resilience is likely to include both qualitative and quantitative components. Some have advocated going further and identifying critical links (Linkov, et al., 2022; Linkov, et al., 2018) or more formally conducting resilience testing (Rogers, et al., 2021). These approaches allow some investigation of the relational issues the authors highlight.

3. Areas of the framework were not discussed in the main text of the article and there may be alternative ways of classifying them or discussing them, for example media as a heading was not discussed in the main text, although trust was. Exactly what is being referred to in the framework was not obvious. For example, did it include or exclude social media. My assumption was that it did.

Very minor comments for the authors, please consider or disregard as appropriate:

- Page 5: line 21, specifically what disciplines are having their knowledge integrated?
- Page 6: Please include the names of the databases searched in the main articles. Should the selection of the databases be justified given the multi-disciplinary approach advocated by the authors?
- Page 25: s the Global Health Index the Global Index Medicus (WHO)?
- I may have missed it but could you include the name or reference the article that you were unable to access in the PRISMA diagram.
- Page 29: Slight inconsistency between the use of population and subpopulation. “at risk subpopulations” and “hard-to-reach populations”
- Page 35 Appendix 3: Consider making it clearer what was the country of the affiliation of the first author and what was the country subject of the review.
- Areas included in the framework that was outlined on page 29 that I thought interesting and perhaps may benefit from consideration included:
 - the specific inclusion of elective in the services that are delayed. Is this necessary? In that is it possible to delay a non-elective service and if so is elective contextual in this case?
 - Recover did not have a specific reference to catching up on delayed and deferred care.
 - The example of crisis standards of care was based around reductions in volume of services than changes in the standards of care, although I assumed that this was in flexible culture and teamwork but it might be worth discussing.
 - Another interesting feature of the framework was relative lack of a legal and regulatory discussion.
 - The discussion and examples (with a few notable exceptions) do not appear to be easily applied to the meta/planetary level.
 - Should the multiple levels be included explicitly in the framework?

- Should the examples be referenced?

The authors highlighted several interesting areas from there synthesis, Five areas that I particularly found interesting and useful were:

1. It is an incredibly ambitious piece of work, interesting reading and the authors are to be congratulated.
2. The formal inclusion of the multiple levels in their framework and highlighting the potential links between them is a useful contribution to the discussion of health system resilience.
3. The explicit inclusion of the patient safety culture as an example was very useful, especially as it was a systems-based approach – which many of the other examples was not.
4. The breakdown of the pre-event time into prepare and prevent. The inclusion of the slow variables to reduce vulnerability was very useful from a conceptual viewpoint, allowing separation of plans from the health of the population.
5. The examples on page 29 were very useful in making the discussion more concrete.

Works Cited

- EU Expert Group on Health Systems Performance Assessment (HSPA). (2020). *Assessing the resilience of health systems in Europe: an overview of the theory, current practice and strategies for improvement*. Publications Office of the EU, Luxembourg. Retrieved 01 29, 2023, from https://ec.europa.eu/health/sites/health/files/systems_performance_assessment/docs/2020_resilience_en.pdf
- Linkov, I., Fox-Lent, C., Read, L., Allen, C., Arnott, J., Bellini, E., . . . Woods, D. (2018). Tiered approach to resilience assessment. *Risk Analysis*, 38(9), 1772-1780. doi:10.1111/risa.12991
- Linkov, I., Trump, B., Trump, J., Pescaroli, G., Hynes, W., Mavrodieva, A., & Panda, A. (2022). *Resilience stress testing for critical infrastructure*. Elsevier Ltd. doi:10.1016/j.ijdr.2022.103323
- Rogers, H., Barros, P., De Maeseneer, J., Lehtonen, L., Lionis, C., McKee, M., . . . Kringos, D. (2021). Resilience testing of health systems: How can it be done? *International Journal of Environmental Research and Public Health*, 18(9). doi:10.3390/ijerph18094742

VERSION 1 – AUTHOR RESPONSE

Reviewer 1	This is a valuable meta-narrative review and synthesis of reviews on an important topic. It is well done and reported, transparent about decisions, and worthy of publication	Thank you for this comment.
	It would possibly be improved by the authors' being a bit clearer about exactly how their synthetic definition arose and how it relates to the very different approaches they identified (e.g. engineering, emergency management)- what is left out in this synthesis and does it matter?	We realised that the conduct of this synthesis was placed in different places in the manuscript, which made it vague. It has now been summarised in the paragraph stating our synthesised definition. Essentially, this was a result of word occurrence analysis across reviews (which included definitions from different disciplines), tiered coding into themes, and further refinement based on the analysis of concepts of resilience. There was strong convergence of the concepts of resilience and the definition that emerged. Some other aspects that were not included tended to deal with the engineering paradigm (e.g. the point at which a material breaks).
Reviewer 2	This is a well-designed and executed review of an important topic. The authors have done a fine job in reviewing the literature and have used interesting methods in doing so--e.g. using a world clad to look at the most-used concepts in relation to healthcare resilience. The choice of focussing on organisational and system levels is well-explained and justifiable given the wealth of the literature on the topic. The definition of resilience the authors derive from the literature is also well argued. I specifically liked that the authors not only analyse and summarise the literature but also come up with suggestions for further research and feel that the turn the authors describe in the literature from more structural to more processual approaches to resilience is a valid one.	These comments are much appreciated, thank you.
	The authors rightly point out that a limitation of the method they chose is that it remains rather abstract, while they point out that resilience is largely to be found in the nitty-gritty details of the relationality between layers (e.g. micro-meso-macro relations). I feel they could elaborate a bit more on how this relationally might look like and what kind of work would be needed to research this. For example, Wiig et al have	Thank you for pointing out Wiig and her team, whose study protocols came up during our screening process. We had not included this as their study is still ongoing and the publication was for the study protocol, but recognise its importance. We have now included a reference to their study in the manuscript.

	pointed at the importance of 'reflexive spaces' between layers whereas others have pointed at the role of 'dynamic protocols'.	
Reviewer 3	I enjoyed reading the paper and think it is a valuable contribution to a very fast-moving field. The authors are to be congratulated on a synthesis that was clear and highlighted several important issues. I also thought their proposed framework highlights several important issues that are valuable to conceptualising resilience.	Thank you very much for the compliments
	Please include more formally the inclusion and exclusion criteria in the methods. One area in which I got confused was the issue of community resilience.	Thank you. We have done this. The main paper on community resilience is what you identified from NZ's Living Standards Framework. We included the framework as it was particularly useful for considering meta-level. It was not actually included in the analysis. We have removed it from the framework summary for clarity.
	Because the review was not restricted with regard to the income of countries, this may have had implications for countries with larger voluntary or informal sectors.	Yes this is a good point. Some countries have robust data about voluntary sectors and unpaid care, while others have little data. Nevertheless, even in high income countries, such as UK, almost 10% of the population are engaged in unpaid caring roles. In the past this has been somewhat underestimated, but the UK's Health and Care Act 2022 tries to formalise some of these relational aspects outside of the formal healthcare sector. Internationally, better appreciation about the role of voluntary organisations and unpaid care may help to expose further nuances and lessons with regards to healthcare resilience.
	Additionally, although it can be inferred from appendix 3 it may be worth noting the coverage of high income health systems versus low-middle income health systems. The definition of exactly what is in the health care sector may vary depending on the context.	The above comment also touches on this point, and challenges the at times arbitrary boundaries of what is considered healthcare.
	I would have appreciated more information about the quality assessment results. As discussed in page 7 the quality varied greatly and some guidance on what could be improved might be useful for future authors of reviews.	Thank you for highlighting this. We have now elaborated on the quality assessment results. Another statement is also added toward the end of the manuscript to highlight the need for more robust reviews in this field.

	Some of the weaknesses that may be produced from the methods used could have been made more explicit. The lack of inclusion of the grey literature is entirely justifiable from the pre-specified methods but does leave some potential information from national and international organisations excluded, for example European Commission’s work on resilience (EU Expert Group on Health Systems Performance Assessment (HSPA), 2020), similarly with the restriction on English. It may be these are more important for the global level of the framework.	We have added in a few sentences discussing this, acknowledging the balance struck between practicality and thoroughness. As with the point about HIC/LMIC selection, the potential language issue is perhaps partially mitigated by the inclusion of many international studies even if reviews are in English only. As you mention, grey literature contains such a large volume of resilience documents that it would have been highly impractical to include them. We have prioritised peer reviewed and high quality reviews. This will hopefully make our findings robust, reproducible, and valid.
	Similarly, it would be worth commenting on the completeness of search strategy and the approach used to make it complete. One specific issue noted was the use of “organisation” in quotes for the Pubmed search, alternatives could have been “organization” or the combination of “organisation” or “organization”. See the table below for the potential differences in the number of recovered articles for screening. A similar issue may occur with “crisis” and “crises”.	Thank you for this. We did include alternative spellings and have included this statement in the methods. The large number of duplicates suggests that the synonyms we used encompassed most of the suitable literature. In addition, hand-searching and snowball sampling also identified reviews we may have potentially missed through the search. With regards to the specific question about "organisation" vs "organization" and "crisis" vs "crises", other databases used automatically included both (Scopus). Scopus drew the most number of hits, so it is likely that both spellings were captured in the search.
	Section 3.1: The definition of healthcare resilience was: Our definition of healthcare resilience is the ability of healthcare workers, organisations or systems to (a) prepare for and prevent, (b) absorb and adapt to maintain structure and function, (c) recover and review from crises, shocks or stressors. Please consider the following issues:	
	Does essential need to be included with function? Or is it all functions?	Most reviews mention essential functions yes. We have changed this to reflect this nuance. We also hope that the discussion around crisis standards or care and changing of function stimulates deeper consideration about this point.
	Is the meta, global system discussed in the article accurately described as a healthcare system? It is not inaccurate but important components of the global health architecture sit outside of health and different considerations are made at a meta level than a national or regional one (https://apps.who.int/gb/ebwha/pdf_files/WHA75/A75_20-en.pdf).	Thank you for this point. There are international interdependencies when it comes to wider determinants of health. A key part of this has been highlighted by the Lancet Commission into Climate Change as an independent determinant of health. Yet, LMICs tend to suffer more health effects from climate change than HICs. Part of the point of including meta-level is to reframe health as not just formal healthcare and explicitly include the wider determinants of health. While such determinants, organisations and architectures may sit

		outside of healthcare, they affect the health of populations and therefore warrant consideration. However, the focus of this review of reviews is not to go into detail about these, but to draw attention to the interdependencies.
	Section 3.3. Measurement of resilience was a key component of the paper and extremely interesting to read. The lack of inclusion of empirical studies may result in validation studies not being included but this is not an important issue given the prespecified search strategy. There appeared to be a potential opportunity to discuss the association between measurement and the framework the authors have designed. Specifically, while the levels concept was included there was less inclusion of the time dimension of the framework (specifically the pre, intra and post). More explicit linking of this section to the framework that was developed would be useful. There were some specific areas in the themes presented on page 29 were this could have been undertaken.	Thank you. Please let us know if we're interpreting this correctly: you feel that the measurement section could have signposted to time-domains that the framework used? If this is the case, perhaps the reflections on Linkov's work below may be satisfactory? We have now included several sentences specifically to link the measures with the time-domain of the framework, and use this as a signpost for appendix 4 that contains other examples of resilience frameworks. Thank you. We have also included a short discussion about the limited availability of measurement tools and indicators in the post- and trans-event domains. In terms of time-domain on the framework, this was included in the framework (both in the diagram and the table). Perhaps it wasn't as obvious on the table-form. We have now colour-coded the framework in the table-form to match that of the diagram. It would be very difficult to include the levels into the framework without creating a 3-dimensional diagram. We are sorry if we have misunderstood your point
	As the authors have discussed measuring resilience is likely to include both qualitative and quantitative components. Some have advocated going further and identifying critical links (Linkov, et al., 2022; Linkov, et al., 2018) or more formally conducting resilience testing (Rogers, et al., 2021). These approaches allow some investigation of the relational issues the authors highlight	Thank you for this extremely valuable insight. We had not come across this, but it does seem highly relevant to resilience assessment. However, there remain several concerns, particularly with regards to healthcare. The first is the potential for increasingly complex and overly detailed sets of indicators which may not necessarily equate to better resilience (https://www.ncbi.nlm.nih.gov/pmc/articles/PMC2645155/). Linkov's tiered approach does not fully explore this. Secondly, similar tiered approaches have been used in infectious disease modelling and hospital infrastructure indices, but have been of limited use during the COVID19 pandemic. Thirdly, the ability to map the healthcare system and its interorganisational components may not necessarily equate to the ability to make changes across organisational boundaries. It's not known

		how best we can work across such boundaries. Nevertheless, the tiered approach a useful concept. Perhaps most importantly is the use of multiple assessment tools for different purposes, at individual tiers. We have added in a paragraph to this effect, and have also signposted to current studies into the relational aspects of resilience.
	Areas of the framework were not discussed in the main text of the article and there may be alternative ways of classifying them or discussing them, for example media as a heading was not discussed in the main text, although trust was. Exactly what is being referred to in the framework was not obvious. For example, did it include or exclude social media. My assumption was that it did.	Thank you. Due to the word limits, we had to restrict the results to a handful of concepts. We decided to focus on more general concepts that could be applied in many areas, and that challenge previous understanding of the topic. The full framework was created from concepts gleaned from the reviews. The inclusion of social media was important particularly during the COVID19 pandemic, but since that review discussed it in detail, we left it out of our own discussion. We agree that there will be many other ways of classifying the concepts. However, the inclusion of these considerations was felt to be more important than the exact means of categorising.
Reviewer 3 minor comments	Page 5: line 21, specifically what disciplines are having their knowledge integrated?	Thank you. The "these" was redundant and now reads "across disciplines".
	Page 6: Please include the names of the databases searched in the main articles. Should the selection of the databases be justified given the multi-disciplinary approach advocated by the authors?	Yes this has been added.
	Page 25: s the Global Health Index the Global Index Medicus (WHO)?	Yes, this has been changed.
	I may have missed it but could you include the name or reference the article that you were unable to access in the PRISMA diagram.	https://wmpilc.org/ojs/index.php/ajdm/article/view/128 We were not able to access this article, but in addition, it was also was not specific to resilience but only to redevelopment. We have changed the PRISMA diagram to reflect this
	Page 29: Slight inconsistency between the use of population and subpopulation. "at risk sub- populations" and "hard-to-reach populations"	Thank you for spotting this inconsistency. It has been changed

	Page 35 Appendix 3: Consider making it clearer what was the country of the affiliation of the first author and what was the country subject of the review.	We have changed the affiliation of the 1st author to the first column
	Areas included in the framework that was outlined on page 29 that I thought interesting and perhaps may benefit from consideration included:	
	the specific inclusion of elective in the services that are delayed. Is this necessary? In that is it possible to delay a non-elective service and if so is elective contextual in this case?	The traditional division between elective and emergency services help managers plan for contingencies. The delay or reduction in services is necessary more systems that operate with thinner margins of capacity (e.g. over 85% bed occupancy rates). A reduction in elective activity allows for resources to be made available for the temporary surge in emergency services.
	Recover did not have a specific reference to catching up on delayed and deferred care.	There was little written about this, and the rapidity of recovery only features in 1 review. This is probably because different organisations will recover at different paces and changes to services may be required with large shocks.
	The example of crisis standards of care was based around reductions in volume of services than changes in the standards of care, although I assumed that this was in flexible culture and teamwork but it might be worth discussing.	The use of workarounds and improvisations were important throughout the COVID19 pandemic and come under flexible culture and teamwork. As you said, the Crisis Standards of Care refer to reduction in volume of services. We have changed this statement to be clearer
	Another interesting feature of the framework was relative lack of a legal and regulatory discussion.	Yes thank you. There are ongoing issues about governance structures when working across organisational boundaries. One examples is the current focus on integrated care systems in the UK (and the historical understanding of the related Sustainable and Transformative Partnerships). Helpful as the concept is, actors are finding it difficult to determine the most appropriate ways forward. On the macro and meta-level, issues around accountability of recommendations suggested by the World Health Organisation continue to raise questions about its role. Due to the world limit, and lack of our own expertise, we have chosen not to focus on regulatory structures.
	The discussion and examples (with a few notable exceptions) do not appear to be easily applied to the meta/planetary level.	You are correct, as the meta-level has only recently emerged from increased understanding of climate change and global warming.
	Should the multiple levels be included explicitly in the framework?	This would be nice, but may increase the complexity of the framework, tipping the balance away from useability.
	Should the examples be referenced?	Thank you for this. We felt that in the table form of the framework, referencing all the examples would make the table less useable and in addition, we would require a separate legend to contain all the references, again making the table far larger than it already is.

	The authors highlighted several interesting areas from their synthesis, Five areas that I particularly found interesting and useful were:  1. It is an incredibly ambitious piece of work, interesting reading and the authors are to be congratulated. 2. The formal inclusion of the multiple levels in their framework and highlighting the potential links between them is a useful contribution to the discussion of health system resilience. 3. The explicit inclusion of the patient safety culture as an example was very useful, especially as it was a systems-based approach – which many of the other examples was not. 4. The breakdown of the pre-event time into prepare and prevent. The inclusion of the slow variables to reduce vulnerability was very useful from a conceptual viewpoint, allowing separation of plans from the health of the population. 5. The examples on page 29 were very useful in making the discussion more concrete. 	

VERSION 2 – REVIEW

REVIEWER	Haywood, Philip Centre for Health Economics Research and Evaluation (CHERE), Business I undertake work for international organizations (OECD and European Commission) on health system resilience.
REVIEW RETURNED	16-Aug-2023

GENERAL COMMENTS	Thank you for the opportunity to review this paper for a second time. I have no major comments on this revision. The author appears to have considered and/or incorporated most of my major comments from an initial review, especially in Section 3.1. A stronger linkage in Section 3.3 is now made between measurement and the time-domain of the framework proposed by the author. I still consider that the author could comment on the relevance and limitations of this work for low and middle income countries, appreciating your point that “many reviewed included studies from a range of countries”. I offer the following minor comment, which may potentially improve readability. I found the sequencing and language of sentences on page 4, Section 2 (Search strategy), lines 51-54 a little difficult to follow. All the best with this work and your future endeavors.
---

VERSION 2 – AUTHOR RESPONSE

Reviewer 3 Round 2	I still consider that the author could comment on the relevance and limitations of this work for low and middle income countries, appreciating your point that “many reviewed included studies from a range of countries”.	Almost a quarter of the reviews included were specifically set in LMICs, and most of the others had an international focus. Many of the historical health agendas considered were developed for LMICs. We therefore consider our framework to be relevant to LMICs. For this reason (amongst others), we have refrained for asserting particular priorities for LMICs. We have instead suggested a realist approach to appreciate the contexts of differing organisations and systems. In addition, it is our view that resilience cannot be built just focusing on any particular component. Instead, we advocate for a broad consideration across the range of components. Equally, some of the concepts and lessons from COVID19 were from HICs, but correspond with global health agendas in LMICs. For example, stronger cross-sector networks are required in HICs (highlighted from the COVID19 pandemic), but have been highlighted in historical global health agendas based in LMICs too. We have been mindful to try not to extend the manuscript too much, but have now included some of these points in the updated manuscript and hope that it helps to address some of the concerns around relevance for LMICs.
	I offer the following minor comment, which may potentially improve readability. I found the sequencing and language of sentences on page 4, Section 2 (Search strategy), lines 51-54 a little difficult to follow.	Thank you. We have edited this to read more succinctly.